# The Multifaceted Role of CMA in Glioma: Enemy or Ally?

**DOI:** 10.3390/ijms22042217

**Published:** 2021-02-23

**Authors:** Alessia Lo Dico, Cristina Martelli, Cecilia Diceglie, Luisa Ottobrini

**Affiliations:** 1Department of Pathophysiology and Transplantation, University of Milan, Via F.Cervi 93, Segrate, 20090 Milan, Italy; alessia.lodico@unimi.it (A.L.D.); cristina.martelli@unimi.it (C.M.); cecilia.diceglie@unimi.it (C.D.); 2Institute of Molecular Bioimaging and Physiology, National Research Council (IBFM-CNR), Via F.Cervi 93, Segrate, 20090 Milan, Italy

**Keywords:** PHLPP1, oxidative stress, autophagy, Temozolomide (TMZ), Hypoxia Inducible Factor-1α (HIF-1α), chaperone proteins, therapeutic target

## Abstract

Chaperone-mediated autophagy (CMA) is a catabolic pathway fundamental for cell homeostasis, by which specific damaged or non-essential proteins are degraded. CMA activity has three main levels of regulation. The first regulatory level is based on the targetability of specific proteins possessing a KFERQ-like domain, which can be recognized by specific chaperones and delivered to the lysosomes. Target protein unfolding and translocation into the lysosomal lumen constitutes the second level of CMA regulation and is based on the modulation of Lamp2A multimerization. Finally, the activity of some accessory proteins represents the third regulatory level of CMA activity. CMA’s role in oncology has not been fully clarified covering both pro-survival and pro-death roles in different contexts. Taking all this into account, it is possible to comprehend the actual complexity of both CMA regulation and the cellular consequences of its activity allowing it to be elected as a modulatory and not only catabolic machinery. In this review, the role covered by CMA in oncology is discussed with a focus on its relevance in glioma. Molecular correlates of CMA importance in glioma responsiveness to treatment are described to identify new early efficacy biomarkers and new therapeutic targets to overcome resistance.

## 1. Introduction

In recent years, autophagy, particularly chaperone mediated autophagy (CMA), has gained attention for its involvement in the regulation of cellular redox homeostasis, metabolic reprogramming, senescence, and neoplastic transformation [1,2,3]. Before exploring CMA’s role and the consequences of its deregulation, some premises are needed.

Generally, the term “autophagy” is used to indicate different mechanisms such as macroautophagy, microautophagy, endosomal microautophagy, and CMA [4]. While macroautophagy and microautophagy are nonselective mechanisms, it has been shown that protein degradation by endosomal microautophagy and CMA is limited to specific targets due to the recognition of a specific pentapeptide firstly discovered by Dice J et al. in 1986 [5].

In more detail, CMA belongs to the cell catabolic machinery, involved in protein and organelle degradation in controlled conditions. This machinery includes the multi-enzyme proteasome system and all the different displays of autophagic phenomena. It has been shown that it is a highly specific mechanism behaving as a biosensor of cellular stress and contributing to cell adaptation to external stimuli, such as nutrient deprivation or oxidative stress and cell damage, and that it is finely regulated by multiple mechanisms [6].

CMA can regulate several processes involved in cell metabolic profile, in the maintenance of proteostasis as well as in the regulation of immune response. All these tasks can be related to an anti-oncogenic effect as demonstrated by its specific down-regulation involved in the tumorigenic process [7]. However, after neoplastic transformation, CMA could play a different and multifaceted role which is still under discussion [3,8]. Indeed, in glioma, it has been reported that CMA is involved in resistance to treatments [9], in the degradation of both pro-survival and pro-apoptotic factors [8,10], and in the regulation of immune response [3,11]. Complexity in CMA regulation can at least partially explain the variability encountered in the different experimental settings.

In this review, we aim to explore the different levels of CMA regulation and their implication in determining its role with special attention to glioma and in proposing treatments able to modulate CMA activity in different molecular and cellular contexts.

## 2. The Multi-Level CMA Modulation

CMA belongs to the selective autophagic machineries. Selective autophagy diverges from the other autophagic mechanisms in different respects for the specifically recognized protein domain, the KFERQ (lysine–phenylalanine–glutamic acid–arginine–glutamine)-like motif. This motif is present in proteins that have to be degraded and in the involvement of specific chaperones such as heat shock cognate 70 kDa protein (HSC70 or HSPA8), heat shock protein 90 (HSP90), STIP1 homology and U-box containing protein 1 (CHIP/STUB), as well as lysosome-associated membrane protein 2 (LAMP-2A). As for CMA, the KFERQ motif is recognized and bound by the chaperone HSC70 which, together with its co-chaperones (HSP90, HSP40, BAG3, HSP70-interacting protein Hip and Hop), is responsible for the targeting of the selected protein to the lysosome membrane. The complex made up of the KFERQ-bearing protein and HSC70 interacts with the CMA receptor LAMP-2A on the cytosolic side of the lysosome membrane. This interaction upon LAMP-2A multimerization induces the unfolding of the protein to be degraded and its translocation within the lumen of the lysosome, where it will be degraded by lysosome hydrolytic enzymes [12,13].

During the life of a cell, a complex net of signaling pathways, transcriptional regulators, and post-translational modifications governs all cellular processes. Among these, CMA is characterized by a complex molecular regulation machinery that finely modulates its activity at different levels (Figure 1) [14]. In detail, CMA activity is regulated on one side by modifying protein targetability for their degradation, and on the other by directly modifying the expression and the activity of CMA players. Cells can modulate which proteins can undergo CMA-mediated degradation through the modification of the recognition motifs and of their exposure to other proteins [15]. On the other hand, the modulation of proteins involved in CMA, their correct localization within the cell and their activity, represent further regulatory levels for this lysosomal function [16].

A protein, to be a CMA target, must contain a pentapeptide biochemically correlated to the sequence “KFERQ”, as this motif works as a recognition sequence for chaperone proteins. To be functional, this motif should contain a glutamine (Q) on one side, two positively charged amino acids (lysine K or arginine R), one or two a-polar amino acids (isoleucine I, leucine L, valine V, phenylalanine F) and a negatively charged one (aspartic acid D or glutamic acid E) [17]. A similar motif is present in about 40% of the cellular proteins [17]; however, not all the KFERQ-containing proteins are degraded by CMA. The consensus sequence must be exposed to be recognized by the chaperones, and this does not always happen for all the proteins that contain it, since different cell conditions can foster the exposition of the consensus motif while others make it unattainable by chaperone proteins [18,19]. Another mechanism to transform a protein into a CMA target consists of the post-translational modification of specific sites. A complete KFERQ-like motif is needed for a protein to be recognized by chaperones; however, if a protein possesses only four canonical residues in the motif lacking the negatively charged amino acid, phosphorylation of a near residue (on Serine S, threonine T or tyrosine Y) can compensate for this lack [14]. Similarly, acetylation of a phenylalanine residue makes it analogous to glutamine. Post-translational modification such as ubiquitination or acetylation of a phenylalanine residue can also act as a switch between proteasomal and CMA degradation of the same protein [20,21]. Those modifications could change not only the consensus sequences but also the protein folding, by exposing or hiding KFERQ or KFERQ-like motifs and changing the protein ability to bind chaperones. The variable nature of some proteins as potential CMA targets is evident for the hypoxia-inducible factor (HIF)-1α degradation (whose implication in oncology is discussed below). Indeed, HIF-1α ubiquitination on lysine 63 by the E3 ubiquitin-protein ligase STUB1 [22] fosters its degradation through CMA, while lysine 48 ubiquitination drives this protein to proteasomal degradation [23,24,25].

Another interesting example is the one involving mammalian Ste20-like kinase 1 (MST1; also known as STK4, a protein kinase involved in the pro-apoptotic switch). Its degradation by CMA is dependent on the acetylation of a specific residue that avoids HSC70 binding to the KFERQ domain [26].

These modifications greatly increase the number of potential CMA targets and open new avenues in the modulation of protein subsets that can be degraded by this specific mechanism in different conditions (hypoxia, nutrient availability/starvation, oxidative stress, metabolic stress) or under different stimuli. In this context, other biochemical modifications such as sumoylation, methylation, succinylation, and neddylation can be involved in the protein switch to a CMA target [17].

Finally, the activation or de-activation of transduction signaling pathways could contribute to changing the CMA targetome, also providing a new role for this process that could gain a higher ranking among the cell processes being elected as a cellular regulator of different functions. The wide variety of potential CMA targets confirms the importance of this degradation mechanism and our poor understanding of its real significance and complexity. This complexity could explain, at least in part, the discrepancies found about CMA role in pathologies, especially in cancer [27].

### 2.1. CMA-Related Protein Expression and Activity

CMA activity can be also controlled by the modulation of CMA-related protein availability and activity. The availability of carrier proteins assigned to recognize and bind the KFERQ-like motifs such as HSC70 (and its co-chaperones Hsp40, Hsp90 and Hip) and STUB/CHIP are some examples of second level regulators.

The KFERQ-bearing proteins are recognized in the cytosol by HSC70 in a chaperone complex, shuttling the client protein to the lysosome. HSC70 is also constitutively present in the lysosome membrane, as well as in lysosomal lumen, where it ensures the correct CMA cycle participating at the protein unfolding, translocation [14,15,28], and final degradation by cathepsin [29] while another chaperone, lysHsp90, supports LAMP-2A stability. At this level, it has been reported that lysHSC70 level, instead of the total HSC70 amount in cellular lysates, is proportional to CMA activity [29].

CMA set up and switch on is also strictly dependent on LAMP-2A levels, its multimerization in the lysosomal membrane compartment, and its binding to substrate proteins. LAMP-2A belongs to a group of transmembrane proteins with similar features including LAMP-1 and LAMP-2 which are characterized by different cellular localization. *Lamp-2* is subject to alternative splicing producing three different transcripts resulting in three LAMP-2 isoforms (A, B, and C) which differ in their transmembrane and cytosolic regions and in their localization in the cell. In particular, LAMP-2A is located at the lysosomal level and is the unique isoform implicated in CMA activation; in fact, LAMP-2A binding of KFERQ in the substrate protein is the rate-limiting step for CMA [30]. LAMP-2A expression, half-life and its localization at the lysosomal level are fundamental steps modifying the efficiency of this degradation process. In this regard, all the transcriptional factors or modulators that could affect LAMP-2A expression are critical. It has been recently reported that the transcription factor NFE2L2/NRF2 is involved in the positive modulation of *Lamp-2A* expression [31].

At the functional level, the binding of the complex containing the target protein allows the monomeric LAMP-2A located into the lipid rafts to dissociate from those regions and to be oriented to the fluid membrane area [32]. In this condition, LAMP-2A can undergo a progressive rearrangement of its quaternary structure deeply involved in CMA activity initiation and regulation. LAMP-2A firstly dimerizes and next trimerizes, assembling a channel for the translocation of substrate unfolded proteins within the lysosome [13,33,34]. It has been proposed, for example, that P38MAPK activation is a modulator of the phosphorylation of Threonine 211 and Threonine 213 of LAMP-2A [16]. These biochemical modifications have been described to allow the LAMP-2A multimerization in the lysosome membrane favoring CMA activity [16]. Despite its crucial role, the increase of the lysosomal LAMP-2A is not sufficient for CMA activation. The process is more complex and requires many other players that have to be expressed, to be active and to be specifically located.

Two other proteins involved in the regulation of LAMP-2A multimerization are the glial fibrillary acidic (GFAP) and the elongation factor (EF1)-1α proteins. These factors modify the stability of the multimeric complex and the association with lipid microdomains through a GTP-dependent mechanism. Lysosomal GFAP is present in a phosphorylated form (GFAP-P), resulting from a GTP-dependent reaction, able to bind EF1α, and in an unphosphorylated form (GFAP), which can bind LAMP-2A multimers stabilizing them. However, GFAP-P is also able to bind its unphosphorylated counterpart with even higher affinity, detaching it from LAMP-2A, inducing the disassembly of the multimeric complex, and finally blocking CMA activity. GFAP-P preferentially associates with EF1α, preventing the formation of GFAP/GFAP-P dimers and allowing LAMP-2A to multimerize, but once the protein to be degraded has been translocated inside the lysosome, EF1α changes its conformation, reducing its affinity for GFAP-P [17]. The detachment of GFAP from LAMP-2A leads to the disaggregation of the multimer and the displacement of LAMP-2A in the microdomains where it can be degraded, further regulating CMA activity [35,36].

LAMP-2A importance in the modulation of CMA is also evident by considering the multiple degradative paths it can undergo, being mediated by cathepsins or by metallo- or serine-proteases, whose activity can counteract CMA machinery and can, in turn, be modulated by other processes [37].

Lastly, the availability of specific proteins into the cells could also modify CMA activity on specific targets. Indeed, post-translational regulation can interfere with the choice of the degradative pathway different target proteins go through. For example, STUB/CHIP is an E3 ubiquitin ligase enrolled in the switch between the ubiquitin-proteasome system (UPS) and the lysosome degradation through CMA machinery. It is mainly involved in the ubiquitination of misfolded proteins bound to chaperones: STUB, by interacting with the HSC70, could target KFERQ-bearing proteins to LAMP-2A in the lysosomes. It has been demonstrated that proteins such as HIF-1α, known to be potential CMA targets, need the presence of STUB/CHIP to finalize their CMA degradation [38].

### 2.2. Implications of Accessory Proteins Availability

Accessory proteins, whose activity is not directly ascribed to target protein unfolding, lysosomal translocation, and degradation, are able to modulate CMA activity as the third level of regulation. The two main additional regulators are Pleckstrin homology (PH) domain leucine-rich repeats protein phosphatase 1 (PHLPP1) and the Rac Family Small GTPase (RAC)-1.

PHLPP1 acts as Ser/Thr protein phosphatase. Three isozymes belong to its family: the alternatively spliced PHLPP1α and PHLPP1β, and PHLPP2. PHLPP1 and 2 differ as they act on different AKT isoforms, respectively dephosphorylating AKT 2 and 1, and inhibiting their activity [39,40,41]. PHLPPs can also exert their action on different targets depending on the cellular type and cell condition. PHLPP1 seems to be involved in CMA regulation since it is recruited at the lysosomal membrane in a Rac1-dependent manner. Rac1 once located at the lysosomal membrane contributes to stabilize PHLPP1, acting as a bridge with the lysosome. Here, PHLPP1 can have a direct activity on GFAP-P de-phosphorylation, promoting CMA activity [35]. For all these reasons, it is limiting to take into account only a few functional players in describing CMA activity; many other factors that orbit around CMA-related proteins and modulators could be important to better understand CMA role and regulation.

## 3. CMA in Oncology

In the last few years, CMA has been analyzed in the field of oncology although its role in tumor onset and progression is not yet completely understood. For instance, tumor malignancy seems to be associated with LAMP-2A overexpression with a poor final prognosis [42,43,44] but, at the same time, down-regulation of PHLPP1 has been found in several tumors [45,46]. These observations suggest that CMA’s role in oncology is still controversial and that there is a need to improve our knowledge about this process, especially concerning more efficient cancer treatment. To reinforce CMA’s importance in tumor establishment and progression, among the proteins degraded by CMA there are a wide range of factors possessing different roles in cancer. For instance, some proteins act as tumor promoters (AF1Q, Vav1, PKM2, Eps8, mutant p53, HK2) and other proteins play a role as tumor suppressors (unphosphorylated PED, Rnd3) [15]. Considering all this, even if indirect, it is important to discuss the role of CMA in the modulation of tumor cell survival or death [6].

One condition in which CMA is engaged in a pro-survival role, for example, is during nutrient deprivation. The attempt to counteract the lack of nutrients and growth factors by activating CMA could induce an adaptation to the new availabilities by changing the cell metabolic profile (i.e., degradation of glycolytic enzymes by CMA [47]) and thus promoting cell survival by eliminating pro-apoptotic factors, even if available nutrients for the cells are low [34]. Several studies describe CMA as a key player in delaying apoptosis onset being engaged in the degradation of pro-apoptotic genes such as BBC3 (BCL2 binding component3)/PUMA. However, it is sufficient to phosphorylate BBC3 to avoid its degradation through CMA [10], demonstrating the plasticity of CMA’s role.

It has been also documented that in some cases, CMA activity plays a pro-death role [48,49], which again involves the degradation of glycolytic enzymes and this time can contribute to metabolic catastrophe and cell death [50]. In detail, CMA can degrade enzymes such as hexokinase 2 (HK2) or pyruvate kinase M2 isoform (PKM2), which are involved in the onset and progression of tumors [51]. Furthermore, CMA can also degrade oncogenic proteins such as Mouse double-minute 2 homolog (MDM2) or myc (even if it is not a direct CMA target) reducing the availability of pro-survival factors in the tumor and reducing proliferation boost [52,53]. Finally, the degradation of factors associated with tumor malignancy, such as HIF-1α [54], could address CMA as a pro-death modulator in cancer cells.

## 4. CMA in Glioma

Glioblastoma (GBM) is the most common astrocytic-derived brain tumor in adults, characterized by a poor prognosis mainly due to its resistance to the available treatments. Temozolomide (TMZ) is the gold standard in glioma management. Thanks to its lipophilic nature, TMZ is used as an oral agent efficiently penetrating the blood–brain barrier (BBB) [55]. It is an imidazotetrazine that derives from the alkylating agent dacarbazine. As an alkylating agent, TMZ methylates DNA for almost 90% at the N7 position of guanine and at the N3 position of adenine thus forming N7-methylguanine (N7-meG) and N3-methyladenine (N3-meA), respectively [56]. These complexes are repaired by the base-excision repair (BER) mechanism counteracting the TMZ-mediated cytotoxic effect. Another methylated adduct, O6-methylguanine (O6MG), even if less frequent, is responsible for the main cytotoxic effect of the drug. In fact, O6MG, although its amount is very limited (about 5–10%) compared to N7-meG or N3-meA, induces important DNA damage thanks to its higher stability [57]. During DNA synthesis, the O6MG adducts produce mispairs with thymines, which are recognized by the DNA mismatch repair (MMR) system. MMR can remove the thymine paired with the O6MG, but not the O6MG itself [58], creating a futile cycle of mismatch and repair. This condition supports the formation of DNA single-strand breaks (SSB) and double-strand breaks (DSB) roughly 2–3 cell cycles after TMZ administration. The final consequences of TMZ administration are then the appearance of stalled replication forks with damage to the DNA [59], the activation of the checkpoint kinase 1 and 2 (Chk1 and Chk2), the consequent inactivation of the cell division cycle 25 (cdc25), the cell cycle arrest and the accumulation of cells at the G2/M boundary [60,61]. Finally, all these events cause the induction of cell death through the activation of apoptosis, senescence, and mitotic catastrophe [58,62].

TMZ cytotoxic effect can be counteracted by the presence of the enzyme O6-methylguanine-DNA methyltransferase (MGMT) that can remove the methylation from O6MG within the DNA in a single step. This reaction is irreversible because when the methyl group is transferred from guanine to the active cysteine residue of MGMT, the protein becomes non-functional and is degraded [63,64].

However, TMZ also exerts other effects in the cells, reflecting the whole toxic activity of this drug. It has been described that TMZ supports the induction of autophagy and endoplasmic reticulum (ER) stress, inducing mitochondrial membrane depolarization and finally releasing reactive oxygen species (ROS) in the cytoplasm [8,65,66]. ROS release in the cytoplasm has been associated with the induction of CMA activity [8], an event that is observed after TMZ treatment (Figure 2).

It is now interesting to reconstruct a model bringing together some observations about the TMZ effect, CMA activity, and cellular consequences schematically reported also in Table 1. It is well known that TMZ can block the cell cycle through Chk1 activation [60,61]. Phosphorylated Chk1 can, in turn, interact with HSC70 for its specific degradation through CMA machinery [67] that can be activated by TMZ. Moreover, CMA involvement in glioma responsiveness to TMZ has been recently reported by Lo Dico et al. [8], demonstrating that CMA engagement, mediating responsiveness to TMZ treatment, is strictly related to a transitory ROS release from mitochondria. In detail, in TMZ-responsive cells, CMA activation after treatment is crucial for the cytotoxic effect to be exerted, since, in resistant cells, TMZ fails in triggering CMA. Only the re-activation of CMA activity allowed the recovery of cell responsiveness to this drug [8].

CMA involvement in TMZ’s mechanism of action has been also proposed by Shi et al. [68], describing LAMP-2A stabilization after treatment with retinoic acid (RA). Indeed, in glioma, it has been shown that RA can support TMZ’s cytotoxic effect by improving autophagic induction and blocking cell proliferation in U251 glioma cells [68].

However, the relationship between autophagy and pro-death mechanisms is not yet clear and is unique, since often it is mediated by other mechanisms.

Recent evidence shows that several tumors, including glioma, express high LAMP-2A levels, and this condition has been associated with a worse prognosis [69]. In GBM cells, a combined positivity to lysosomal LAMP-2A and intraluminal Cathepsin D has been described, even if the expression level of CMA-related genes was different among the analyzed cell lines (U87 vs. T98 glioma cells). In the same work, it was also shown that LAMP-2A was detected in 20% of the neoplastic cells [70], although no clues about the actual CMA activity in those models have been provided. A recent paper showed that the treatment with lovastatin was able to down-regulate LAMP2 and dynein in U251 and U87 cells. Dynein is indeed involved in the trafficking of the autophagosome and its abrogation is related to CMA impairment since it is also enrolled in the localization of LAMP-2A at the lysosomal level [71]. Thus, lovastatin might exert its effect by the down-regulation of both LAMP-2 and dynein, showing that these molecules could become molecular targets to counteract the CMA switch on [72].

At the same time, a down-regulation of PHLPP1 in several tumors has been described, reversing the supportive hypothesis correlating CMA activity and tumors [73,74] and suggesting that the role of its activation in tumors is still not fully understood.

To our knowledge, there is no evidence in the literature of a complete analysis on the whole pathway involving and regulating CMA activity, and probably this is the reason for the inconsistent findings already reported and for the need for further studies about this issue.

Other CMA-related proteins have been associated with glioma aggressiveness. Among those proteins, HSC70 is not only commonly expressed in glioma, but it has also been related to glioma grade [75]. A recent study showed that its down-regulation was correlated with the reduction of cell viability and proliferation in glioma models [76]. However, the meaning of these results needs to be fully understood and confirmed. Indeed, in our work, the abrogation of HSC70 expression in TMZ-responsive glioma cells made them resistant to this treatment [8]. As a possibility, heterogeneity and variance of HSC70′s role in different cells and conditions cannot be excluded, enhancing the importance of further studies regarding CMA-related proteins in glioma resistance or responsivity to treatments.

In line with the PHLPP1 decrease in tumors, our evidence showed also that within the CMA pathway a key role is played by PHLPP1 both as a molecular supporter of LAMP-2A-driven CMA and as a potential biomarker. Indeed, a TMZ-responsive model (such as U251 cells) was characterized showing a CMA-network in which *Lamp-2A*, *Hsc70*, and *Phlpp1* expression were significantly up-regulated by TMZ treatment. The opposite molecular profile has been indeed observed in a TMZ-resistant model (such as T98 cells). This molecular pattern thus supports the activation of CMA as a process that mediates the responsiveness to TMZ-based chemotherapy [8].

Looking at the multicellular system composing glioma, recently the key role of pericytes (PC) in the progression of GBM and in the efficacy of chemotherapy was described [3]. PCs are perivascular stromal cells known to drive neurovascular processes and to maintain the efficiency of the blood–brain barrier, thus also influencing the result of chemotherapy in GBM patients [77]. The same study by Valdor et al. [3] showed the role of pericytes in GBM also in the switch from a pro-inflammatory to an anti-inflammatory phenotype. In this transition, CMA covers a fundamental role. In physiological conditions, PCs promote inflammation in response to cell damage. On the other hand, during GBM progression, PCs play an immunosuppressive role, through the onset of an anti-inflammatory phenotype, which finally induces the immune tolerance against tumor cells and allows their proliferation [3].

This evidence suggests that the immune response could modulate tumor outcome through specific modulation of CMA. In another paper, it was reported that the oncosuppressor PHLPP1, known to have an anti-proliferative action on tumor cells [39,78], is also involved in the activity of T cells and immune response. Its specific ability to dephosphorylate Akt plays a role in regulating T cell activity by negatively modulating regulatory T cells (Treg) [79,80,81]. Tregs possess an immune-suppressive role, and they can down-regulate the activation of effector T cells: by increasing PHLPP1 activity, this suppressive role is lost, and the establishment of an anti-tumoral microenvironment is favored [11]. Supported by these results, it could be assumed that CMA is activated in T cells [81].

However, also in the context of immunity, the role of PHLPP1 is not clear. Beyond its role in regulating Treg, it controls the inflammatory response by also acting on the dephosphorylation of Ser127 of the Signal transducer and activator of transcription (STAT)-1. This dephosphorylation inhibits the activity of the transcription factor, blocking the consequent expression of different mediators of innate immunity and cytokines [11] that normally permit the triggering of the innate immune system against infections.

Considering together the role of PHLPP1 in the activation of CMA and in the modulation of inflammatory stimuli, this factor can be elected as a predictive biomarker of responsiveness to standard therapy in glioma [46].

All these considerations pave the way for the inclusion of CMA activity in the modulation of molecules involved in the immune response. However, in glioma, this process is not still completely elucidated and further studies should be carried out.

## 5. CMA Molecular Correlates with Glioma Resistance and Responsiveness to TMZ

CMA blockage has been described as a driver for resistance to TMZ mainly because of its role in the modulation of several factors involved in glucose metabolism and in the regulation of transcription, translation, and cell cycle control [6]. Among them, HIF-1α, one of the factors associated with glioma malignancy, can be also degraded by CMA [54]. The switch on of hypoxia in glioma is of great importance not only for the survival of cells and for the acquisition of an aggressive phenotype, but also for the onset of tumor resistance to standard therapy [84,85]. HIF-1α, indeed, is the main driver of cell response to hypoxia and other critical processes enrolled in glioma transformation and proliferation such as angiogenesis, epithelial to mesenchymal transition, invasiveness, and stemness [86,87]. For these reasons, HIF-1α has been implicated in glioma resistance to treatments due to its role in the inhibition of cell apoptosis and in the promotion of mechanisms that finally support cell survival [9]. This is why HIF-1α degradation through CMA has been related to responsiveness to treatment in GBM [9]. In detail, HIF-1α can be degraded by two distinct processes within the cells: the first mechanism is mediated by proteasome activity and oxygen-dependent HIF-1α ubiquitination machinery mainly regulated by Von Hippel-Lindau (VHL) [88]. The second mechanism is oxygen-independent and is driven by CMA. HIF-1α contains the KFERQ domain and through STUB/CHIP activity it can be delivered to the lysosome for its CMA-mediated degradation [22]. The role of CMA activity in HIF-1α degradation is particularly evident after the down-regulation of LAMP-2A or HSC70: the silencing of even one of the genes encoding these proteins, crucial for CMA activity, is sufficient to increase HIF-1α level and activity in the cells [8]. As a confirmation, a higher expression of these master genes driving CMA is related to an increase in CMA-mediated HIF-1α degradation [54]. Similarly, all the pharmacological treatments designed to block the CMA process (i.e., bafilomycin [82] or chloroquine [83], even if they are a-specific for CMA inhibition compared to the molecular silencing) allow the accumulation of this transcriptional factor, increasing the expression of the vascular endothelial growth factor (VEGF), proliferation, glucose metabolism, and finally cell survival [89]. On the contrary, the overexpression of transcription factor EB (TFEB) [70], enrolled in lysosomal biogenesis, is involved in the decrease of HIF-1α level and activity [54] and all pharmacological treatments aimed at increasing CMA activity (digoxin, for example) allow the decrease of HIF-1α cell availability and function [54,90].

In our recent paper [9], we showed a reciprocal ratio between *Hif-1α* and *Lamp-2A* expression in TMZ-responsive cells that is completely inverted in resistant ones. These data are in line with TMZ-mediated CMA activation and the consequent HIF-1α degradation in TMZ-responsive cells. All these considerations contributed to elect HIF-1α not only as an early biomarker of responsiveness to TMZ [9], but also as a biomarker of CMA switch on/off in glioma, being crucial for glioma cell responsiveness to TMZ treatment.

All these results support the key role of HIF-1α in mediating GBM resistance and show how the specific up-regulation of CMA, which allows HIF-1α reduction in the cells, is of extreme importance in GBM management and response to treatment.

## 6. Concluding Remarks: CMA as a Therapeutic Target

In light of these considerations and evidence, it is not so clear if CMA constitutes a potential target to be blocked by a treatment or a powerful ally to be used to increase the potency of the treatment itself or to overcome resistance. Probably, several variables can tip the balance in favor of one or the other role, depending on cell condition, nutrient availability, and presence of stress stimuli. Indeed, at the basal level, CMA is active for the maintenance of cellular homeostasis [91]. However, its inhibition is needed in specific conditions, for example, to reduce the degradation of the pro-apoptotic factor BBC3 [10]. This kind of approach could be associated with radiotherapeutic and chemotherapeutic strategies, able to add genotoxic damage that finally provokes apoptotic cell death.

On the other hand, it has been also documented that enhanced CMA can possess a pro-death potential [48,49]. This is what happens, for example, by using a combined treatment based on mammalian target of rapamycin (mTOR) inhibitor and TMZ in GBM: this treatment sustains the induction of the autophagic onset, favoring cell death. This is a new strategy in which the induction of other pro-death inputs could overcome the intrinsic resistance to standard therapy.

In our hands, this evidence has been also confirmed showing that not only mTOR inhibitors but also generic treatments able to increase intracellular ROS can overcome resistance to TMZ in glioma cells by inducing CMA activity [8,9].

Herein, the controversial role of CMA in glioma has been discussed, reporting some examples of how CMA modulation can become a new potential therapeutic target for cancer treatment. Deeper studies should be carried out to elucidate which tumors or cell conditions can benefit from CMA activation or inhibition. However, given the several emerging publications and the progressive improvement of knowledge about this autophagic mechanism, it is clear that CMA can justifiably be elected as a new dual-purpose tool, being both an important responsiveness biomarker and an interesting new therapeutic target in the field of precision medicine.

## Figures and Tables

**Figure 1 ijms-22-02217-f001:**
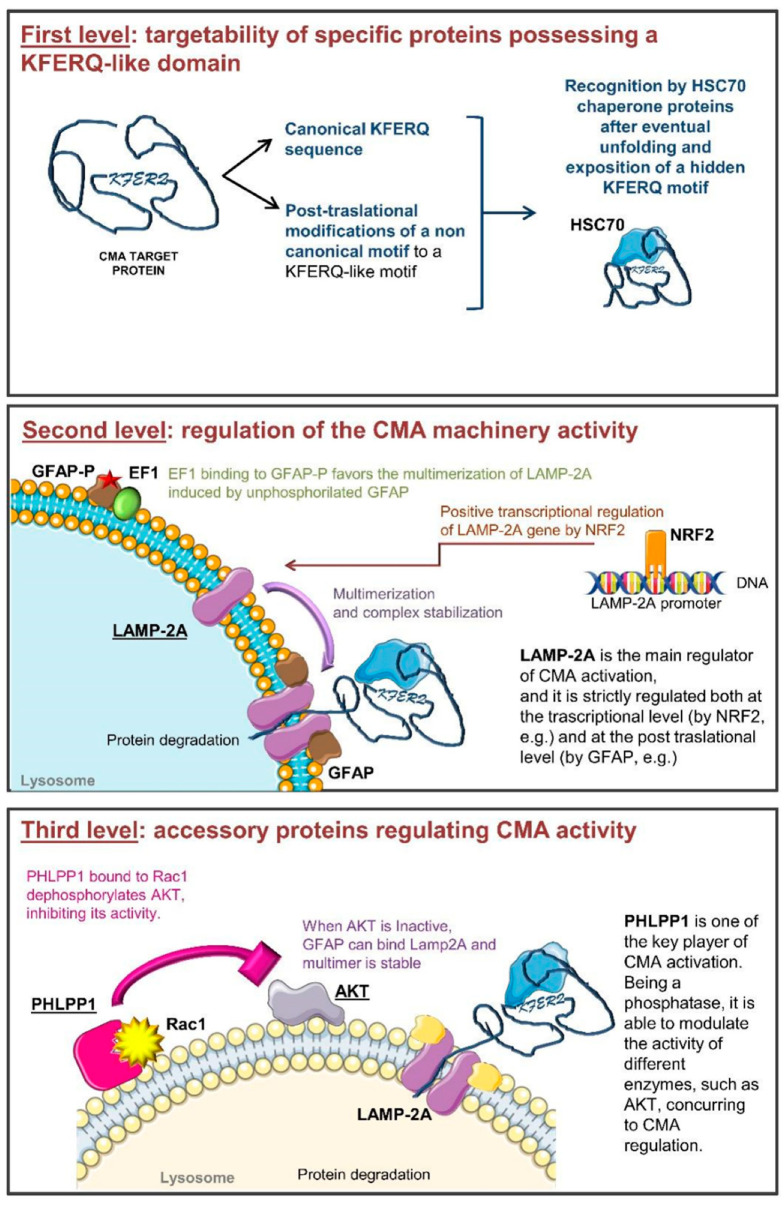
Schematic representation of the multi-level CMA (chaperone-mediated autophagy) regulation machinery. **Panel 1:** The first regulatory level is based on the availability and accessibility of the KFERQ motif. Non-canonical motifs can be modified to be recognized by chaperone proteins. HSC70 chaperone delivers specific proteins to the lysosome. **Panel 2:** The second regulatory level depends on LAMP-2A quaternary structure. Its trafficking in the lysosomal membrane is finely regulated. Moreover, unphosphorylated GFAP (GFAP) stabilizes the LAMP-2A multimer, favoring the translocation of unfolded proteins within the lysosomes. When GFAP is present in its phosphorylated form (GFAP-P), it interacts with EF-1α, determining the disassembling of LAMP-2A multimers. In addition, LAMP-2A de novo expression can be regulated by the transcription factor NRF-2. **Panel 3:** The third regulatory level is due to the availability of accessory proteins among which the main regulator is PHLPP1. It is a phosphatase influencing the activity of other proteins. Rac-1 bound to PHLPP1 favors its activity on AKT. Inactive AKT is not able to phosphorylate GFAP allowing LAMP-2A multimer stabilization.

**Figure 2 ijms-22-02217-f002:**
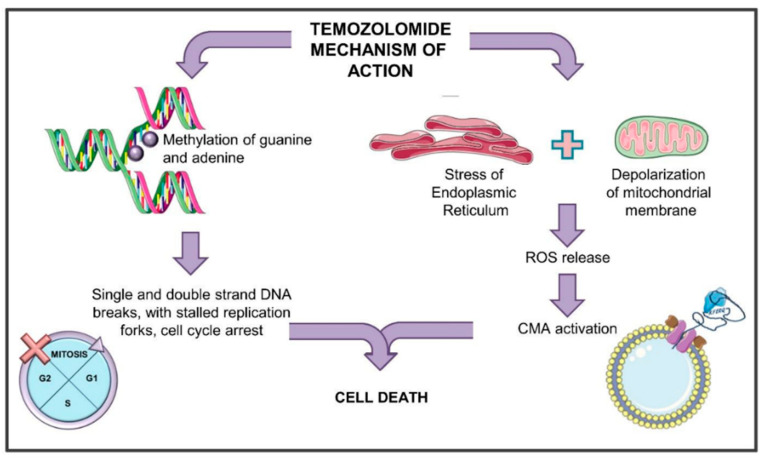
General scheme of the mechanism of action of temozolomide. TMZ (temozolomide) can induce different events. On one hand, it methylates DNA forming N7-methylguanine (N7-meG) and N3-methyladenine (N3-meA), and less frequently, O6-methylguanine. DNA methylation induces the DNA mismatch repair (MMR) system that cannot remove the O6MG, creating futile cycles of mismatch and repair and generating DNA single- and double-strand breaks. Finally, stalled replication forks appear, and the cell cycle is blocked. All these events induce cell death through the activation of apoptosis, senescence, and mitotic catastrophe. On the other hand, TMZ can induce endoplasmic reticulum stress with consequent mitochondrial membrane depolarization and release of reactive oxygen species (ROS) in the cytoplasm. ROS release is associated with the induction of CMA activity, which in turn concurs to cell death.

**Table 1 ijms-22-02217-t001:** Description of the CMA activities and regulations involved in glioma features and responsiveness to treatments.

	Key Player	Relation with Glioma	References
**CMA regulation in glioma**	**Dynein**	It is involved in autophagosome trafficking and LAMP-2A localization. It has been targeted with therapeutic purposes to block CMA.	[71]
	**TFEB**	Highly expressed in glioblastoma cells responsive to TMZ.	[54]
	**LAMP-2A**	LAMP-2A high level correlates with a worse prognosis.	[69]
	**HSC70**	Commonly expressed in glioma, lysosomal HSC70 is related to glioma grade.	[8,75]
	**PHLPP1**	PHLPP1 down-regulation has been found in several tumors including glioma.	[8,73,74]
**CMA targets in glioma**	**Chk-1**	TMZ treatment induces Chk1 phosphorylation and consequent degradation through CMA.	[67]
	**HIF-1α**	HIF-1α reduction in glioma cells has been related to TMZ treatment efficacy.	[8,9,22,54]
	**Other CMA target**	Several tumor suppressors and oncogenes involved in tumorigenesis are CMA targets (such as mutant p53, BBC3/PUMA, HK2, etc) but still need further investigation in glioma.	[10,15,50,51]
**CMA and glioma treatments**	**TMZ**	Autophagy inducer: ROS-mediated induction of CMA activity in relation to responsiveness to the treatment.	[8,65,66]
	**ROS**	CMA inducer: oxidative stress helps in overcoming TMZ resistance.	[8]
	**RA**	CMA inducer: LAMP2A stabilization; it supports the cytotoxic effect of TMZ treatment.	[68]
	**Digoxin**	CMA inducer: reduces glioma growth and increases tumour sensitivity to anticancer treatments.	[54]
	**Lovastatine**	Autophagy inhibitor: reduces LAMP-2A and dynein levels. Used with TMZ to enhance its efficacy.	[71,72]
	**Bafilomycin**	Autophagy a-specific inhibitor: reduces cancer stem cell viability and increases HIF-1α level.	[54,82]
	**Chloroquine**	Autophagy a-specific inhibitor: adjuvant for RT, its role is still controversial in different models and conditions.	[54,83]
**CMA activity and immune response in glioma**	**T regs**	PHLPP1 induces negative modulation of Treg and the consequent increase of specific adaptive immune response against glioma.	[39,78]
	**Machropages**	PHLPP1 blocks STAT-1 activity, inhibiting innate immune response.	[11,79,81]
	**Perycites**	CMA activation induces anti-inflammatory microenvironment and glioma immune-tolerance.	[3]

## Data Availability

Not applicable.

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
