# Peer review of "The Multifaceted Role of CMA in Glioma: Enemy or Ally?"

_ijms, 2021, doi:10.3390/ijms22042217_

Round 1
Reviewer 1 Report
The authors present a review which focuses on the role of chaperone mediated autophagy in glioma. Mechanism of action and its regulations are extensively described. Furthermore, the role in gliomas with an emphasis on glioblastoma is delineated and the option of targeting this mechanism therapeutically is discussed. Overall, the manuscript is interesting to read and suitable for this journal. Written English is readable, but consultation of a native speaker would make the manuscript more appealing to the reader.
Introduction:
Becomes very specific at the end. It would be more fitting to give a general introduction in the field of CMA and why it is important to keep reading this review (relevance in the field of oncology and glioma, possible targetability). Furthermore, large parts of the introduction are lacking references.
The multi-level CMA modulation:
Again, many statements are not backed up by literature.
Both figures are of low quality and the written words are very hard to read.
Author Response
Reviewer 1
The authors thank the reviewer for the suggestions.
We performed a linguistic revision and we hope that now the manuscript will be more readable.
For the introduction, we revised the text including only concepts while the more specific information have been included in the following paragraph.
In general we add more references in whole text. We hope the reviewer will agree with our changes.
As regards figures, we have improved the resolution.
We hope reviewer will agree with our revision and will recommend the manuscript for publication.
The authors are available to provide any other clarification or to perform further changes if needed.
Reviewer 2 Report
This is a comprehensive review on an original link between a selective type of autophagy: chaperone mediated autophagy and oncology, with an emphasis on glioma. This collected work at the forefront of knowledge connecting a basic and a translational field will be useful for both cell biologist and oncologists. In general, the text is well organized and clear. Though, most of the links between CMA substrate/effectors and cancer are indirect and this should be stated clearly in the manuscript. Also, a table compiling the key references linking glioma with CMA substrates/regulators would be helpful for readership.
Other points: -line 124: please introduce "lysHSC70"; -line133: pelase correct the type "...in fact..."; -lines 144-153: please introduce the reference describing these mechanisms; -line244: please correct the type "...to its..."; -line381: please correct "...they are not specific...".
Author Response
The authors thank the reviewer for the suggestions.
Reviewer 2
We thank the reviewer for the comments and the advices.
We have added a table (see attachment) including all the references related to glioma and CMA. We hope the reviewer will agree with it.
For the other points:
-line 124 we have introduced lysHSC70
- line 133 the authors have corrected the typo “in fact”.
- lines 144-153 we added all the references describing the mechanisms
- lines 244 and 381 we corrected the mistakes.
We are grateful to the reviewer for the comment The authors are available to provide any other clarification or to perform further changes if needed.
We performed a linguistic revision and we hope that now the manuscript will be more readable.
For the introduction, we revised the text including only concepts while the more specific information have been included in the following paragraph.
In general we add more references in whole text. We hope the reviewer will agree with our changes.
As regards figures, we have improved the resolution.
We hope reviewer will agree with our revision and will recommend the manuscript for publication.
The authors are available to provide any other clarification or to perform further changes if needed.
